# DISTILLING REASONING INTO STUDENT LLMS: LO-CAL NATURALNESS FOR SELECTING TEACHER DATA

## ABSTRACT

Distilling long reasoning traces (10K+ tokens) from stronger teacher models into smaller student LLMs via supervised fine-tuning (SFT) has emerged as a standard paradigm. This approach is both practical and efficient: it leverages the ease of generating abundant reasoning data from stronger models and provides a direct, data-driven way to teach less capable models better reasoning. While previous work has largely focused on prompt selection with responses from a single teacher, the equally important problem of choosing the best response when multiple teacher outputs are available for a single prompt remains underexplored. This challenge becomes especially important in a multi-teacher setting, where different students may benefit from the outputs of different teachers. This paper fills that gap with a systematic study of response selection for reasoning distillation. We first show that the current method, which picks the response that the student assigns the highest global log-probability (i.e., global "naturalness"), fails when responses come from multiple teachers. In such cases, global naturalness no longer correlates with downstream performance, especially as the reasoning traces from strong teachers become longer. To overcome this limitation, we introduce Local Naturalness, which scores a response by measuring the student's log-probabilities over short, sequential reasoning steps (e.g., sentences) conditioned only on a small local window. Local Naturalness enables two novel applications: 1) Teacher Selection: Aggregating local scores across prompts reliably identifies the most helpful teacher, whereas global scoring fails completely. 2) Response Selection from a Mixed-Teacher Dataset: When mixing answers from many teachers, Local Naturalness boosts a 32-billion-parameter student's accuracy on math benchmarks by $9.4\%$ over global-naturalness-based selection, also surpassing the performance achieved by training on data from the single best teacher. These results highlight the power of localized data quality evaluation and data mixing for more effective reasoning distillation.

## 1 INTRODUCTION

Large language models (LLMs) have reached the point where reasoning, not merely fluent text generation, has become the next frontier. Supervised fine-tuning (SFT) on long chain-of-thought (CoT) (Liu et al., 2024) exemplars distilled from stronger reasoning teacher models, such as DeepSeek-R1, Qwen3, or QWQ-32B, is now the workhorse for pushing a student model toward more sophisticated, multi-step reasoning (Guha et al., 2025; Ye et al., 2025; Liu et al., 2025; Shen et al., 2025).

Recognizing the fundamental role of data in modern machine learning, recent research has increasingly focused on data selection strategies. Most efforts, however, stop at the prompt level, such as curating prompts for diversity or difficulty (Muennighoff et al., 2025; Ye et al., 2025; Zeng et al., 2025), and implicitly assume that each prompt has a single, fixed teacher response. In reality, researchers and practitioners often have access to multiple teachers, each capable of producing many distinct responses to the same prompt. Those responses could vary in logical depth, clarity, and alignment with the student's current knowledge, so keeping the "right" response that better fits the student could matter as much as choosing the right prompt (Shen et al., 2025). Yet student-aware response selection remains underexplored.

Zhang et al. (2025) introduce first principal attempts at response selection. Responses to which the student assigns the highest *global* log-probability during next-token prediction are retained. The

intuition is that data the model already finds "natural" will be easiest to learn. While the heuristic works well when all candidate responses come from a *single* teacher with shorter CoT responses, in multi-teacher settings with long reasoning data (10K+ tokens), however, it breaks down: the student's global scores do *not* generalize across teachers with long reasoning data—a response with a lower global likelihood can still yield superior downstream accuracy.

**From global to local assessment.** We hypothesize that global log probability is an unreliable metric for evaluating performance on long-context tasks. This is because student models, which were trained with shorter context windows, often struggle to maintain information consistency when processing extended input lengths (10,000-30,000 tokens), a form of degradation that global metrics can fail to capture (Liu et al., 2023). Driven by this insight, we propose Local Naturalness: instead of a single global log-probability for the entire response, we measure the log-probabilities of successive reasoning steps, each conditioned on a limited preceding window.

We evaluate Local Naturalness across two critical applications: firstly, the selection of an optimal teacher model, and secondly, the curation of superior responses from a heterogeneous dataset generated by multiple teachers. Our experiments demonstrate that Local Naturalness robustly identifies the most effective teacher model. Furthermore, when selecting individual responses, it consistently assembles a subset of data that yields the best downstream performance across diverse student models. Notably, this local selection strategy significantly outperforms existing global scoring method and can achieve results surpassing those obtained by training on the dataset from the best-performing individual teacher model.

**Contributions.** This paper makes the following contributions:

- We provide the first systematic analysis showing that global log-probability is an unreliable indicator of training utility in mixed-teacher settings.

- We introduce Local Naturalness, a simple scoring rule that leverages the student's own token probabilities at the sentence level.

- Across student sizes from 7 B to 32 B and teachers including Qwen 3.0, QWQ, and DeepSeek-R1, our method consistently improves reasoning performance on math and science benchmarks.

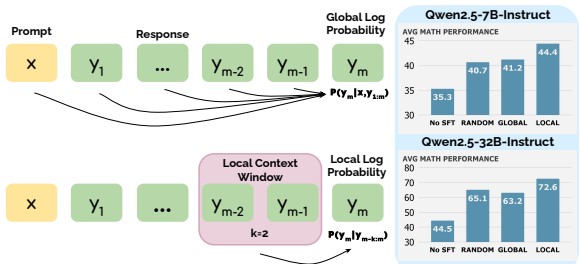

Figure 1: We contrast two data selection metrics: the global log-probability, computed with the entire preceding context, and the local log-probability, computed within a local sliding window. Model performance is reported both before and after SFT when training data are selected by three strategies—Random sampling, Global scoring, and our proposed Local scoring.

More broadly, this work contributes to the development of model-aware data curation techniques, an increasingly critical direction as LLMs are pushed to tackle longer and more cognitively demanding reasoning tasks.

## 2 RELATED WORK

The task of curating effective data for SFT of LLMs is an active and critical area of research. High-quality SFT data is essential for adapting pre-trained models to specific downstream tasks and aligning them with desired behaviors. Our work builds upon and differentiates itself from several existing lines of inquiry, particularly in the generation and use of synthetic reasoning data, model-aware data selection, and knowledge distillation.

**Synthetic instruction data curation.** The use of LLMs to generate synthetic reasoning data, especially long chain-of-thought (CoT) responses (Wei et al., 2022), has become prevalent for tasks requiring multi-step inference. For instance, Guo et al. (2025); Ye et al. (2025); Liu et al. (2025), have demonstrated methods for generating such data. The generation of synthetic data also involves processes of critique and revision, sometimes employing other AI models as evaluators to filter or refine the generated examples, ensuring higher quality and alignment. However, as Guha et al. (2025) observed, it is not necessarily a stronger teacher that is always more beneficial for a student model.

**Model-aware data selection.** Recognizing that a one-size-fits-all approach to SFT data is sub-optimal (Li et al., 2025), researchers have explored various model-aware data selection strategies. GRAPE (Zhang et al., 2025) made a significant step by proposing to select SFT data based on the student model's global generation (log) probability of the entire response. This aims to select responses "natural" to the student model's pretraining distribution. Our work directly builds on GRAPE, acknowledging its strengths for selecting responses from a single teacher but identifying its limitations for mixed-teacher settings and long reasoning data. Beyond global probabilities, some methods involve offline student modeling, where a surrogate student-response predictor is trained to score potential training examples ((Kostrikov et al., 2021; Bai et al., 2021)). While potentially effective, these often require expensive offline training loops. Another avenue is online curriculum learning, which dynamically adapts sampling probabilities during training ((Liang et al., 2021; Lu & Zhang, 2021)), though this can necessitate modifications to the training process itself. Active learning strategies aim to select the most informative data points for labeling or fine-tuning, thereby reducing annotation costs and improving model efficiency. For instance, SIFT (Selection by Information-theoretic Fine-Tuning) (Hübotter et al., 2024) combines retrieval and active learning to select data that reduces uncertainty about the model's response. Influence functions have also been explored to identify training samples that have the most impact on a model's predictions or specific validation samples, although their application to LLMs presents challenges due to scalability and convergence issues (Choe et al., 2024). Our method, by focusing on inherent model probabilities, offers a simpler, more direct way to achieve model-awareness for reasoning data without auxiliary models or complex training loop modifications.

**Knowledge distillation.** Our approach shares conceptual similarities with knowledge distillation (KD), where the goal is to transfer knowledge from a (typically larger) teacher model to a smaller student model. Classical KD aligns the two models at *token level* by minimizing the KL divergence between their output distributions at every decoding step (Gou et al., 2021; Song et al., 2025). Although effective, this step-wise alignment is computationally expensive because the teacher must be run in lock-step with the student throughout training. A more efficient alternative is *response-level* KD: a teacher first generates complete responses, and the student is later trained on those sequences with ordinary cross-entropy loss (Hsieh et al., 2023; Gupta et al., 2023). Our method belongs to this family but adds a *student-aware filter*.

## 3 METHOD

### 3.1 PROBLEM DEFINITION: RESPONSE SELECTION

In this section, we provide necessary notations, define the problem of *response selection* for supervised fine-tuning (SFT) of LLMs, and introduce the concept of global log probabilities.

**Notations.** We denote a sample as an input-output pair $(x, y)$, where $\mathbf{x} = (x_1, \ldots, x_n)$ represents the input prompt sequence, and $\mathbf{y} = (y_1, \ldots, y_m)$ represents the output response sequence. For a given student model $S$ with parameters $\theta_S$, the generation probability of a response $y$ conditioned on the input prompt $\mathbf{x}$ is expressed as $P(y|x; \theta_S)$. The average log probability of the response $y$ is then defined as the average of the log probabilities of each token in the response, given the input prompt and the previously generated tokens:

$$\overline{\log}P(y|x; \theta_S) = \sum_{t=1}^{m} \frac{1}{m} \log P(y_t|y_{1:t-1}, x; \theta_S),  \tag{1}$$

where $y_{1:t-1} = (y_1, \ldots, y_{t-1})$ denotes the sequence of tokens generated up to the $(t-1)^{th}$ token (with $y_{1:0}$ being an empty sequence). This log probability quantifies how "natural" or likely the model considers the response $y$ given the input prompt $x$. We term $\overline{\log}P(y|x; \theta_S)$ as the *global log probability* of the response $y$ given the input prompt $x$, as it considers the entire response sequence conditioned on the input.

**Problem statement.** Given a set of candidate responses $Y = \{y^{(1)}, y^{(2)}, \ldots, y^{(k)}\}$ generated from diverse sources for a specific input prompt $x$, our goal is to select the most suitable response $y^* \in Y$ for SFT of a student $S$ using the pair $(x, y^*)$. In this paper, we primarily focus on the mathematical reasoning domain. This domain provides access to long, structured reasoning data and allows for straightforward verification of response correctness. Throughout the paper, we evaluate model performance on a wide suite of math benchmarks, including MATH-500, AIME 2025, AMC,

MINERVA, KAOYAN, OLYMPIADBENCH, CN_MATH_2024, and we evaluate based on accuracy. Please refer to the Appendix for further dataset details and for LiveCodeBench evaluation.

## 3.2 RE-EXAMINING GLOBAL LOG PROBABILITY WITH MIXED TEACHERS

**Global Log Probabilities.** As proposed by Zhang et al. (Zhang et al., 2025) (referred to as GRAPE), one method for response selection is to choose the response with the highest global log probability as calculated by the student model $S$. For a given prompt $x$ and a set of candidate responses $Y$, the selected response $y^*$ is:

$$y^* = \arg \max_{y \in Y} \overline{\log} P(y|x; \theta_S).$$

This approach assumes that the response most "natural" to the student model (i.e., having the highest log probability according to its current parameters) is the most informative for SFT, thereby leading to better downstream performance. While the original work demonstrated GRAPE's effectiveness on general domain data, we aim to investigate its applicability specifically to reasoning data, considering multiple teacher models.

**Task Setup.** To evaluate the effectiveness of GRAPE, we conduct initial experiments using MATH prompts of level 3-5 difficulty, resulting in 8890 prompts. For each prompt, we generate 16 responses using a teacher model with temperature 0.6 and 0.95 top-p sampling, employing vLLM (Kwon et al., 2023) for generation. We ensure that the final answer in each response matches the ground truth. Using the GRAPE method, we select the response with the highest global log probability for SFT. The student model is then fine-tuned on these selected responses for 5 epochs with a learning rate of 1e-5 and an effective batch size of 64. For comparison, we also train the student model on responses with the lowest and middle log probabilities from the set of 16 candidates. We conduct these experiments using two teacher: Qwen2.5-72B-Instruct (Yang et al., 2024b) and Gemma-27B-IT (Team et al., 2025), and two student models: Qwen2.5-7B-Instruct and a domain-adapted variant, Qwen2.5-Math-7B. We evaluate using greedy decoding in this task.

**Preliminary Observations.** Our results, detailed in Table 1 and illustrated in Figure 2, student models trained on responses with the highest log probability generally achieve the best performance compared to those trained on responses with lower or middle log probabilities. Furthermore, Figure 2 indicates a positive correlation between the average performance of the student model and the average global log probability of the training data within the same teacher model. However, the results do not hold across teachers. For example, as we observe in Figure 2, in Qwen-7B-Instruct, the responses with the lowest global log probabilities from the Qwen2.5-72B-Instruct teacher responses yield the lowest model performance (0.292), while with these global scores, the responses from Gemma3-27B-IT reach much higher performance (0.313). Moreover, for the Qwen-Math-7B model, we also observe that the two teacher lines do not align, showing some divergence between teachers. These observations motivate us to further investigate into whether global naturalness can be effective for selecting responses that come from even more capable reasoning teacher models.

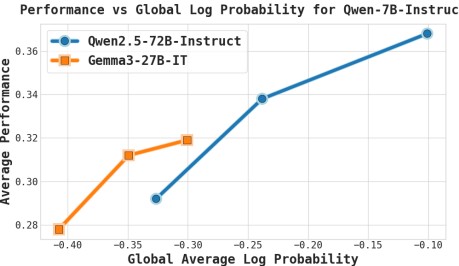 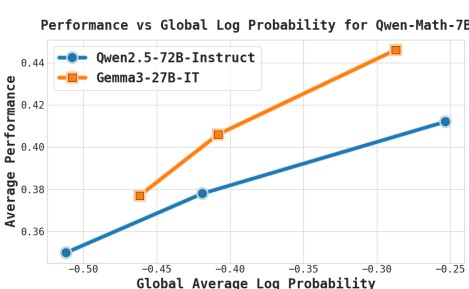

Figure 2: Average Performance vs Global Log Probabilities (scaled by $10^2$) of the Data used for Model Training on Teacher Data (Left) Qwen-7B-Instruct as a Student (Right) Qwen-Math-7B as a student for short data.

## 3.3 LIMITATIONS OF GLOBAL LOG PROBABILITIES FOR LONG REASONING DATA

The demonstrated value of reasoning data in enhancing the capabilities of leading LLMs, such as OpenAI's O-series (Jaech et al., 2024), DeepSeek-R1 (Guo et al., 2025), Nemotron (Bercovich et al.,

|  |  | MATH | AIME25 | AMC | MINERVA | KAOYAN | OLYMPIADB | CN_MATH24 | AVG |
|---|---|---|---|---|---|---|---|---|---|
| | | | | | Student: Qwen2.5-7B-Instruct | | | | |
| | Original Model | 0.752 | 0.167 | 0.5 | 0.268 | 0.216 | 0.404 | 0.167 | 0.353 |
| **Teacher:** | Lowest LP | 0.678 | 0.1 | 0.3 | 0.224 | 0.296 | 0.314 | 0.133 | 0.292 |
| **Qwen2.5-72B** | Middle LP | 0.71 | 0.1 | 0.425 | 0.257 | 0.336 | 0.339 | 0.2 | 0.338 |
| **-Instruct** | Highest LP | 0.744 | 0.133 | 0.5 | 0.252 | 0.391 | 0.391 | 0.167 | 0.368 |
| **Teacher:** | Lowest LP | 0.667 | 0.1 | 0.375 | 0.165 | 0.226 | 0.29 | 0.133 | 0.279 |
| **Gemma3-27B** | Middle LP | 0.716 | 0.1 | 0.475 | 0.129 | 0.246 | 0.357 | 0.167 | 0.313 |
| **-IT** | Highest LP | 0.712 | 0.1 | 0.5 | 0.176 | 0.251 | 0.362 | 0.133 | 0.319 |
| | | | | | Student: Qwen2.5-Math-7B | | | | |
| | Original Model | 0.5 | 0.033 | 0.425 | 0.092 | 0.1 | 0.164 | 0.133 | 0.207 |
| **Teacher:** | Lowest LP | 0.77 | 0.033 | 0.5 | 0.26 | 0.407 | 0.381 | 0.1 | 0.350 |
| **Qwen2.5-72B** | Middle LP | 0.79 | 0.1 | 0.55 | 0.25 | 0.41 | 0.416 | 0.133 | 0.378 |
| **-Instruct** | Highest LP | 0.778 | 0.133 | 0.6 | 0.35 | 0.46 | 0.398 | 0.167 | 0.412 |
| **Teacher:** | Lowest LP | 0.802 | 0.133 | 0.525 | 0.213 | 0.331 | 0.436 | 0.2 | 0.377 |
| **Gemma3-27B** | Middle LP | 0.792 | 0.1 | 0.575 | 0.246 | 0.312 | 0.45 | 0.367 | 0.406 |
| **-IT** | Highest LP | 0.816 | 0.167 | 0.625 | 0.25 | 0.387 | 0.455 | 0.433 | 0.448 |

Table 1: Performance of Qwen2.5-7B-Instruct and Qwen2.5-Math-7B student models on various reasoning tasks when fine-tuned with responses from different teacher models using MATH prompts. The log probabilities (LP) are categorized into lowest, middle, and highest based on the generation log probabilities of the student model.

2025), or QWQ-32B (Team, 2025c), is significant. Moreover, recent work has successfully distilled long, complex reasoning abilities (10K+ tokens) from these highly capable teacher models to smaller student models (Ye et al., 2025; Team, 2025a; Wen et al., 2025). These advancements underscore the importance of effective data selection strategies. This motivates our critical examination of methods like GRAPE, specifically for their efficacy in selecting high-quality, long-form reasoning data. We posit that the difficulty in preserving information consistency across long contexts (Liu et al., 2023) undermines the reliability of global log probability as an evaluation metric. This issue is particularly pronounced in student models not exposed to long-chain reasoning (10,000-32,000 tokens) during their training, suggesting that a metric measuring across the entire sequence may overlook the potential inconsistency of the student model.

**Task Setup.** To investigate this, we conduct experiments using 817 mathematical prompts, filtered according to LIMO (Ye et al., 2025), which have shown to successfully elicit long reasoning chains and improve performance of student models. We generate responses using three distinct teacher models known for their reasoning capabilities: Qwen3-32B-Instruct (Yang et al., 2025), QWQ-32B, and DeepSeek-R1 (Guo et al., 2025). For generation, we use a temperature of 0.6 and top-p sampling of 0.95 with vLLM (Kwon et al., 2023), continuing to sample until the response's final answer matches the ground truth. For each set of teacher-generated data, we fine-tune two student

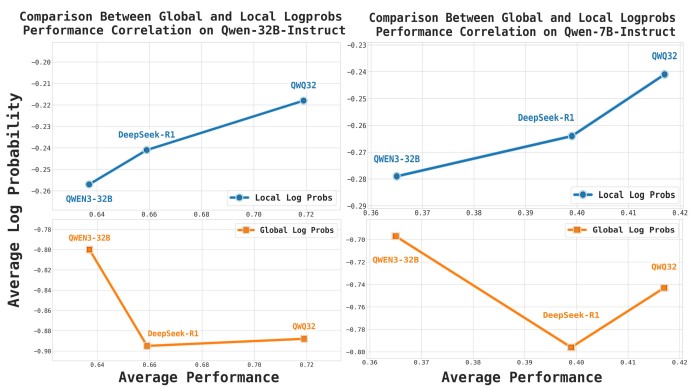

Figure 3: Average Performance vs Global and Local Log Probabilities (scaled by $10^2$) of the Long Reasoning Data from three Teacher models used for Model Training on Teacher LIMO Data (Left) Qwen-32B-Instruct as a Student (Right) Qwen-7B-Instruct as a student.

models: Qwen2.5-7B-Instruct and Qwen2.5-32B-Instruct. The fine-tuning follows LIMO's recommendations: 10 epochs for the 7B model and 15 epochs for the 32B model, with learning rates of $1 \times 10^{-5}$ and $5 \times 10^{-6}$, respectively. We compute the global log probabilities of the training responses using the *pre-fine-tuned* student models to assess their initial perception of the data's "naturalness," consistent with the GRAPE methodology.

**Observation and Motivation.** Figure 3 reveals a crucial discrepancy. Student models trained on responses generated by Qwen3-32B achieved the lowest performance among the three teacher datasets, despite these Qwen3-32B responses having the highest average global log probability as assessed by the student models. Conversely, responses from QWQ-32B, which did not yield the highest global log probabilities, led to the best student model performance after fine-tuning.

This inconsistency strongly suggests that for reasoning data, global log probabilities (the core metric of GRAPE) may not be a reliable indicator of training effectiveness. The student model's global "naturalness" assessment of a long chain of thought does not consistently correlate with the actual performance.

Consequently, relying on these global scores can obscure high-quality reasoning segments embedded within longer responses. Overcoming the unreliability of global log probabilities assigned by student models to long reasoning data is therefore a central challenge. This motivates our exploration of a more localized approach to data selection, focusing on the "naturalness" of individual reasoning steps rather than the entire response, to better identify data that genuinely enhances student model capabilities.

## 3.4 METHODOLOGY: LOCAL LOG PROBABILITIES

The insight from (Prystawski et al., 2023) that effective reasoning, like chain-of-thought, often arises from a "locality of experience", where models learn to chain accurate local inferences, is pivotal. Their work suggests that robust step-by-step reasoning depends on the model's proficiency with local statistical dependencies. Building on this and addressing the limitations of global log probabilities for reasoning data discussed in Section 3.3, we introduce an alternative data selection strategy.

Instead of relying on the student model's evaluation of an entire, extended response (which can be unreliable for less specialized or overwhelmed models), our approach focuses on assessing the generation probabilities of smaller, constituent logical steps within that response. This "local" assessment aligns more closely with the constructive nature of reasoning. Individual logical steps, being shorter and less complex, are more amenable to accurate probability estimation by student models that might falter with the full sequence's length and intricacies. By evaluating these "stepping stones" of the reasoning process, we aim to directly assess the quality and suitability of the intermediate components for the student model. This shift to local log probabilities (exemplified in Figure 1) leverages the model's ability to comprehend and evaluate more manageable logical units, thereby offering a more reliable measure of reasoning quality in long responses across different teacher models and mitigating the biases inherent in global log probability assessments.

**Definition.** We define the **local log probability** of a response $y$ as the averages of the log probabilities of its constituent logical steps, where each step is conditioned on a limited context of $k$ preceding logical steps. Formally, let a response $y$ be composed of $p$ logical steps, $y = (s_1, s_2, \ldots, s_p)$, where each step $s_i$ is a sequence of tokens. The local log probability of $y$ given an input prompt $x$ is:

$$\overline{\log}P_{\text{local}}(y|x; \theta_S) = \frac{1}{p}\sum_{i=1}^{p} \overline{\log}P(s_i|s_{\max(1,i-k):i-1}, x; \theta_S), \tag{2}$$

where $s_{\max(1,i-k):i-1}$ represents the sequence of at most $k$ logical steps immediately preceding step $s_i$. If $i \leq k$, the context includes all preceding steps $s_1, \ldots, s_{i-1}$ and the initial prompt $x$. When $i = 1$, $s_1$ is conditioned only on $x$. This formulation captures the model's assessment of each logical step within a localized context, allowing for a more granular evaluation of reasoning quality. In our practical implementation, we define these logical steps $s_i$ as individual sentences within the response.

**Experimental Setup.** To understand the relationship and divergence between local and global log probabilities, we conduct an analysis. We use the LIMO prompts and the corresponding reasoning responses generated by our three teacher models (DeepSeek-R1, Qwen3-32B, QWQ-32). For these responses, we compute their local log probabilities using a student model with varying context window sizes $k$ for the logical steps (sentences). Specifically, we test context sizes corresponding to approximately 5%, 25%, 50%, and 75% of the total number of sentences in the preceding part of the response. We then compare these local log probability rankings with the ranking derived from the standard global log probabilities (Equation 1) for the same set of responses.

**Observation.** As illustrated in Figure 4, several key patterns emerge: local log probabilities calculated with smaller context windows (i.e., 5% and 25% of preceding sentences) are generally significantly higher than global log probabilities. This suggests that the student model expresses greater confidence (assigns higher probability) when evaluating individual logical steps within a limited, recent context, compared to evaluating the entire, potentially very long, response. Morever, we observe that the ranking of teachers remains stable at smaller context window size (<25%). For smaller context windows, the ranking of teacher data based on local log probabilities can differ substantially from the global log probability ranking. For instance, with

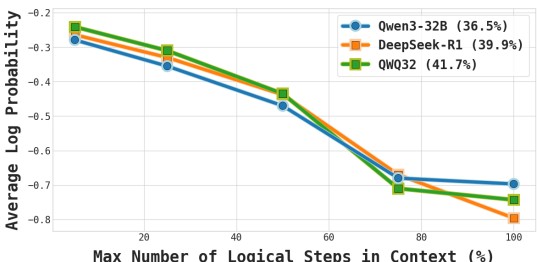

Figure 4: Average log probabilities (scaled by $10^2$) with increasing context window showing the convergence to the global log probabilities ranking. Qwen-7B-Instruct as a student model trained with LIMO response from the teacher (avg SFT performance reported).

a small context, QWQ-32Bresponses might consistently rank highest locally. However, as the context window size $k$ increases (e.g., towards 50% or more of the preceding sentences), the local log probabilities and their induced rankings begin to converge towards those of the global log probabilities. At larger context sizes, the ranking might flip, aligning with the global ranking where, for example, Qwen3-32B responses had the highest global scores but poorest downstream performance (as noted in Section 3.3). This divergence at smaller context windows shows that local log probabilities offer a fundamentally different, and potentially more nuanced and reliable, assessment of reasoning quality than global scores, especially when the student model is still developing its long-context reasoning capabilities. The ability of local scores to highlight "naturalness" at the step level, independent of the global assessment of the entire chain, forms the basis for our proposed response selection method. Consequently, we will explore the efficacy of using these local log probabilities, particularly those derived from shorter contexts, for selecting long reasoning responses across teachers for SFT in the subsequent sections.

## 4 EXPERIMENTS

In this section, we present experiments designed to evaluate the efficacy of our proposed local log probability-based method for selecting long reasoning responses. We compare its performance against selection based on global log probabilities. Our investigation focuses on two primary scenarios:

**1. Teacher Model Selection for a Given Student:** Identifying the most suitable teacher model (and its generated data) for fine-tuning a specific student model.

**2. Cross-Teacher Data Selection for a Given Student:** Selecting the best individual responses from a pool generated by multiple different teacher models for fine-tuning a specific student model.

### 4.1 STUDENT-AWARE TEACHER MODEL SELECTION

The choice of teacher model can significantly impact a student model's fine-tuning outcome, as the alignment between the teacher's data characteristics and the student's pre-existing knowledge (pretraining distribution) varies. Identifying an optimal teacher is therefore crucial. This experiment investigates whether local log probabilities, as assessed by the student model, can effectively guide the selection of the most suitable teacher model for generating long reasoning data.

**Experimental Setup.** We employ Llama-3.1-8B-Instruct, Qwen2.5-7B-Instruct, Qwen2.5-32B-Instruct as our student models. The teacher models are Qwen3-32B-Instruct, DeepSeek-R1 (Guo et al., 2025), and QWQ-32. For each teacher, we generate responses to the LIMO prompts (Ye et al., 2025). Each student model is then fine-tuned separately on the full set of responses generated by each respective teacher model. The fine-tuning hyperparameters (epochs, learning rates) are consistent with those described in Section 3.3 (i.e., LIMO's recommendations). For results on other training datasets (8890 prompts), we refer the reader to Appendix B. Due to limited space, we show all results on Llama-3.1-8B-Instruct model in Appendix B.

After data generation but *before* fine-tuning, we compute both the global and local log probabilities of all generated responses using the specific student model that will be trained on that data. For local log probabilities, we use a context window of at most $k = 4$ preceding sentences, the size we found to be optimal for performance and efficiency. For ablation study, we refer the reader to Appendix. The goal is to see if the average local log probability for a teacher's dataset, as perceived by a student, correlates with that student's post-fine-tuning performance.

| | MATH | AIME25 | AMC | MINERVA | KAOYAN | OLYMPIADB | CN_MATH24 | AVG | Global LP | Local LP |
|---|---|---|---|---|---|---|---|---|---|---|
| **Student: Qwen2.5-7B-Instruct** | | | | | | | | | | |
| **Student Before SFT** | 0.752 | 0.167 | 0.500 | 0.268 | 0.216 | 0.404 | 0.167 | 0.353 | - | - |
| **Qwen3-32B Data** | 0.714 | 0.166 | 0.500 | 0.279 | 0.389 | 0.375 | 0.133 | 0.365 | -0.697 | -0.279 |
| **DeepSeek-R1 Data** | 0.784 | 0.166 | 0.600 | 0.239 | 0.330 | 0.441 | 0.233 | 0.399 | -0.796 | -0.264 |
| **QWQ-32BData** | 0.780 | 0.266 | 0.600 | 0.275 | 0.356 | 0.442 | 0.2 | 0.417 | -0.743 | -0.241 |
| **Student: Qwen2.5-32B-Instruct** | | | | | | | | | | |
| **Student Before SFT** | 0.822 | 0.133 | 0.700 | 0.298 | 0.422 | 0.471 | 0.233 | 0.445 | - | - |
| **Qwen3-32B Data** | 0.882 | 0.567 | 0.900 | 0.353 | 0.598 | 0.559 | 0.600 | 0.637 | -0.800 | -0.257 |
| **DeepSeek-R1 Data** | 0.896 | 0.467 | 0.925 | 0.338 | 0.613 | 0.644 | 0.733 | 0.659 | -0.895 | -0.241 |
| **QWQ-32BData** | 0.916 | 0.633 | 0.975 | 0.364 | 0.653 | 0.689 | 0.800 | 0.719 | -0.888 | -0.218 |

Table 2: Performance of Qwen2.5-7B-Instruct and Qwen2.5-32B-Instruct student models on LIMO prompts when fine-tuned with responses from different teacher models. The log probabilities (LP) are categorized into global and local average log probabilities of the student model.

**Results and Analysis.** The performance outcomes are presented in Table 2, and the relationship between log probabilities and performance is visualized in Figure 3. In particular, student models fine-tuned on QWQ-32B data achieved the highest downstream task performance, the phenomenon which has been observed in Guha et al. (2025) and Xiao et al. (2025). Notably, the QWQ-32B-generated data also exhibited the highest average local log probability when assessed by the student models. This was followed by data from DeepSeek-R1 (second highest local log probability and performance), and then Qwen3-32B data (lowest local log probability and performance). This direct correlation suggests that student-model-assessed local log probabilities are an effective metric for predicting which teacher's data will be most beneficial. In contrast, global log probabilities did not show this clear correlation with student performance in the context of long reasoning data. As previously noted (Section 3.3), Qwen3-32B data had high global scores but led to poorer outcomes. Interestingly, the loss landscape might also be misleading, as models trained with highest global log achieved lowest and fastest convergence (more in Appendix). Computing log probabilities across an entire large dataset can be computationally intensive. We investigated whether a smaller subset of prompts could suffice for reliable teacher model ranking. Our experiments (using 200, 400, and 600 LIMO prompts) demonstrated that local log probabilities derived from as few as 200 prompts were sufficient to reliably rank the teacher models in the same order as when using the full set.

**Takeaways.** These findings indicate that student-model-assessed local log probabilities offer a robust and efficient method for identifying optimal teacher models for generating long reasoning SFT data. This local, "naturalness-at-the-step-level" metric appears better equipped than global scores to capture the true utility of extended reasoning responses, especially when the student (evaluator) model might struggle with full-sequence assessment. The ability to make this selection accurately using a small subset of prompts significantly reduces computational overhead, making the approach practical for large datasets and facilitating quicker identification of suitable data sources.

## 4.2 RESPONSE SELECTION ACROSS TEACHER MODELS

When multiple teacher models are available, they produce data of varying quality and suitability for a specific student model. Effective curation to select the most beneficial individual responses is critical. This experiment investigates the efficacy of using student-model-assessed local log probabilities to select the best individual responses per prompt for a target student model from a diverse pool of outputs generated by different teacher models.

**Experimental Setup.** We use the same student models (Qwen2.5-7B-Instruct, Qwen2.5-32B-Instruct) and teacher models (Qwen3-32B-Instruct, DeepSeek-R1, QWQ-32) as in Use Case 1. For each LIMO prompt, we consider all responses generated by all three teacher models, creating a candidate pool for that prompt. Using the target student model, we compute the **global log probability** and the **local log probability** (with a context of $k = 4$ preceding sentences) for each candidate response. For each prompt, we then create different datasets for fine-tuning by selecting responses based on the following criteria: **Local Highest:** The response with the highest local log probability from the

candidate pool. **Global Highest:** The response with the highest global log probability (GRAPE baseline). **All 3 Teacher:** The 3 responses from all teachers. **Random:** A randomly selected response from the candidate pool (another baseline). The student models are then fine-tuned on these curated datasets using the same hyperparameters as in the previous experiments (Section 4.1). Due to space constraints, we provide results of capable open-weight models, LIMO-32B, Sky-T1-32B-Preview, and OpenThinker2-32B in the Appendix.

**Task Setup.** Similarly as in the previous experiment, we conduct experiments using Qwen2.5-7B-Instruct and Qwen2.5-32B-Instruct as student models and Qwen3-32B-Instruct, DeepSeek-R1, and QWQ-32B as teacher models. We generate responses from each teacher model using LIMO prompts and compute local log probabilities of the responses with a context size of at most 4 previous sentences. For each prompt, we then select the responses with the highest local log probabilities across all teacher models. We also select the responses with the lowest local and highest global log probabilities for comparison. We train the student models on these selected responses with the same hyperparameters as in the previous section. For non greedy decoding with temperature 0.6 and top-p 0.95 with 8 samples for pass@1 evaluation, we refer the reader to the Appendix.

| | MATH | AIME25 | AMC | MINERVA | KAOYAN | OLYMPIAD | CN_MATH24 | AVG |
|---|---|---|---|---|---|---|---|---|
| **Student: Qwen2.5-7B-Instruct** | | | | | | | | |
| Original Model | 0.752 | 0.167 | 0.500 | 0.268 | 0.216 | 0.404 | 0.167 | 0.353 |
| Random | 0.768 | 0.133 | 0.625 | 0.268 | 0.367 | 0.456 | 0.233 | 0.407 |
| Global Highest | 0.762 | 0.2 | 0.6 | 0.268 | 0.381 | 0.441 | 0.233 | 0.412 |
| Local Lowest | 0.742 | 0.167 | 0.575 | 0.298 | 0.342 | 0.433 | 0.233 | 0.399 |
| Local Highest (Ours) | 0.788 | 0.2 | 0.625 | 0.298 | 0.392 | 0.441 | 0.333 | **0.440** |
| **Student: Qwen2.5-32B-Instruct** | | | | | | | | |
| Original Model | 0.824 | 0.133 | 0.700 | 0.298 | 0.422 | 0.471 | 0.233 | 0.445 |
| Random | 0.906 | 0.400 | 0.925 | 0.327 | 0.628 | 0.636 | 0.733 | 0.651 |
| Global Highest | 0.876 | 0.433 | 0.825 | 0.331 | 0.592 | 0.636 | 0.733 | 0.632 |
| Local Lowest | 0.896 | 0.400 | 0.825 | 0.324 | 0.608 | 0.640 | 0.700 | 0.623 |
| Local Highest (Ours) | 0.902 | 0.667 | 1.000 | 0.353 | 0.653 | 0.673 | 0.833 | **0.726** |

Table 3: Performance of Qwen2.5-7B-Instruct and Qwen2.5-32B-Instruct student models on LIMO prompts when fine-tuned with responses from different selection strategies. The log probabilities (LP) are the global and local average log probabilities of the student model.

**Observation.** Table 3 reveals an advantage for using highest local log probabilities in data selection: student models trained on responses chosen this way demonstrated the highest performance, surpassing both random selection and selection via global log probabilities. This underscores the efficacy of local log probabilities for curating high-quality long reasoning responses from different teachers suitable for SFT. The improvement of 0.094 in performance when using highest local versus highest global log probabilities strongly validates our proposed method's effectiveness. A particularly noteworthy finding is that our selection technique can further refine and improve upon the performance of even the top-performing model (Qwen2.5-32B-Instruct trained on QWQ-32B responses from Table 2). This ability to find beneficial individual responses, even from teacher models not deemed globally most helpful, directly addresses the inherent problem of data selection: that true quality can be hidden and not always correlate with a teacher's overall performance. It showcases the nuanced power of our selection criterion and the remarkable effectiveness of our method in precisely identifying these valuable instances.

## 5 CONCLUSION

In this work, we identified a critical limitation of using full-sequence (global) log probabilities to select reasoning examples for supervised fine-tuning across teacher models. Global log probabilities fails at selecting responses from different teacher sources, causing poor correlation between global scores and downstream performance. To address this, we introduced *local log probabilities*, which assess model confidence over individual logical steps within a response. By computing these shorter-context scores, we obtain a more reliable selection criterion that aligns with a student's actual reasoning capabilities. Empirically, local log probability based selection consistently outperforms global log probabilities selection across multiple student-teacher pairs for math reasoning benchmarks. Future work includes adaptive, and informative context window strategies, integrating local scoring into the training loop, and extending beyond mathematical reasoning tasks.

**Reproducibility statement.** The authors have made an effort to ensure the reproducibility of their work. The paper includes a detailed formulation of the method, along with all necessary hyperparameters for data generation, training, and evaluation. Furthermore, all codebases used in this research are properly referenced. In a commitment to fostering further research within the open-source community, the authors will also release the data and models.

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

# A   ADDITIONAL DETAILS

To ensure clarity and facilitate reproducibility, this section outlines the datasets used for training and evaluation, the student model and teacher model architectures, and the hyperparameters used during our supervised fine-tuning process.

## A.1   DATASET DETAILS

**Training Datasets.**   For training, we use two primary datasets. First, we use the MATH dataset (Hendrycks et al., 2021), and following prior works (Zeng et al., 2025; Yu et al., 2025), we filter it to include only questions of difficulty levels 3-5, yielding 8,890 prompts (available at `https://huggingface.co/datasets/EleutherAI/hendrycks_math`). Second, to train models on reasoning data, we use the LIMO dataset (Ye et al., 2025), a carefully curated collection of 817 prompts (available at `https://huggingface.co/datasets/GAIR/LIMO`).

**Evaluation Datasets.**   To evaluate the performance of the model in mathematical capabilities, we include a wide suite of math benchmarks, including:

- MATH500 (Hendrycks et al., 2021) (500 Samples)
  URL:   `https://huggingface.co/datasets/EleutherAI/hendrycks_math`

- AIME 2025 (American Invitational Mathematics Examination) (30 Samples)
  URL: `https://huggingface.co/datasets/opencompass/AIME2025`

- AMC 2023(American Mathematics Competition) (40 Samples)
  URL: `https://huggingface.co/datasets/math-ai/amc23`

- MINERVA (Lewkowycz et al., 2022) (272 Samples)
  URL: `https://huggingface.co/datasets/knoveleng/Minerva-Math`

- KAOYAN (Chinese Graduate School Entrance Examinations) (199 Samples)
  URL:   `https://github.com/GAIR-NLP/LIMO/blob/main/eval/data/kaoyan/test.jsonl`

- OLYMPIADBENCH (He et al., 2024) (675 Samples)
  URL: `https://huggingface.co/datasets/knoveleng/OlympiadBench`

- CN_MATH_2024 (Chinese High School Mathematics League Competition) (30 Samples)
  URL: `https://github.com/GAIR-NLP/LIMO/blob/main/eval/data/cn_math_2024/test.jsonl`

- GPQA-D (A Graduate-Level Google-Proof Q&A Benchmark) (198 Samples)
  URL: `https://huggingface.co/datasets/Idavidrein/gpqa`

- LCBv2 (LiveCodeBench) (511 Samples)
  URL: `https://github.com/LiveCodeBench/LiveCodeBench`

## A.2   MODEL DETAILS

**Student Models.**   For student models, we perform supervised fine-tuning on:

- Qwen2.5-Math-7B (Yang et al., 2024c)
  URL: `https://huggingface.co/Qwen/Qwen2.5-Math-7B`

- Qwen2.5-7B-Instruct (Yang et al., 2024a;b)
  URL: `https://huggingface.co/Qwen/Qwen2.5-7B-Instruct`

- Qwen2.5-32B-Instruct (Yang et al., 2024a;b)

- Llama-3.1-8B-Instruct (Grattafiori et al., 2024)
  URL: `https://huggingface.co/meta-llama/Llama-3.1-8B-Instruct`

**Teacher Models.**    For teacher models, we sample responses from the following models:

- Qwen2.5-72B-Instruct (Yang et al., 2024a;b)
  URL: `https://huggingface.co/Qwen/Qwen2.5-32B-Instruct`
- Gemma3-27B-IT (Team et al., 2025)
  URL: `https://huggingface.co/google/gemma-3-27b-it`
- DeepSeek-R1 (Guo et al., 2025)
  URL: `https://huggingface.co/deepseek-ai/DeepSeek-R1`
- QWQ-32B(Team, 2025c)
  URL: `https://huggingface.co/Qwen/QWQ32b`
- Qwen3-32B (Yang et al., 2025)
  URL: `https://huggingface.co/Qwen/Qwen3-32B`

### A.3  EXPERIMENTAL DETAILS

**Sampling Hyperparameters.**    Training data for fine-tuning student models were generated by sampling outputs from teacher models. We use the vLLM library (Kwon et al., 2023) for this process to ensure efficient inference, employing the sampling hyperparameters detailed in Table 4.

| Property | Value |
|---|---|
| Number of samples | 1/16 |
| Temperature | 0.0/1.0 |
| Top P | 1.0/0.95 |
| Top K | 1/40 |
| Max Tokens | 42786+ |

Table 4: The hyperparameters for sampling from the teacher models using vLLM (Kwon et al., 2023).

**Training Hyperparameters.**    For supervised fine-tuning on student models, we leverage the LLaMA-Factory (Zheng et al., 2024) platform that offers efficient training and apply the following setting of hyperparameters (listed in Table 5):

| Property | Value |
|---|---|
| Train Batch Size Per Device | 1/2 |
| Gradient Accumulation Steps | 8 |
| Learning Rate | $5.0 \times 10^{-6}/1.0 \times 10^{-5}$ |
| Epochs | 10/15 |
| Warmup Ratio | 0.0 |
| BFloat16 | True |

Table 5: The hyperparameters for SFT the student models using LLaMA Factory (Zheng et al., 2024).

**Evaluation Hyperparameters.**    After models are trained, we evaluate the models on a variety of mathemtical benchmarks using the evaluation library from LIMO (Ye et al., 2025) (URL: `https://github.com/GAIR-NLP/LIMO/tree/main/eval`) with the following hyperparameters (Table 6):

After training, we evaluate the models on a range of mathematical benchmarks using the evaluation library provided by LIMO (Ye et al., 2025) based on the Qwen2.5-Math evaluation code (Yang et al., 2024b) (available at `https://github.com/GAIR-NLP/LIMO/tree/main/eval`). The evaluation is conducted using the hyperparameter settings from DeepSeek-R1 Guo et al. (2025) as detailed in Table 6.

| Property | Value |
|---|---|
| Temperature | 0.0/0.6 |
| Max Tokens | 32768 |
| Top P | 1/0.95 |
| Pass@K | 1/8 |
| Samples | 1/8 |

Table 6: The hyperparameters for evaluation of the student models at the inference stage using evaluation code from Qwen2.5-Math evaluation (Yang et al., 2024b).

## B  ADDITIONAL RESULTS

### B.1  LOSS COMPARISON

We compare the loss curves of student models trained on data selected using global and local log likelihood criteria, as summarized in Table 3. The corresponding loss plots are presented in Figure 5.

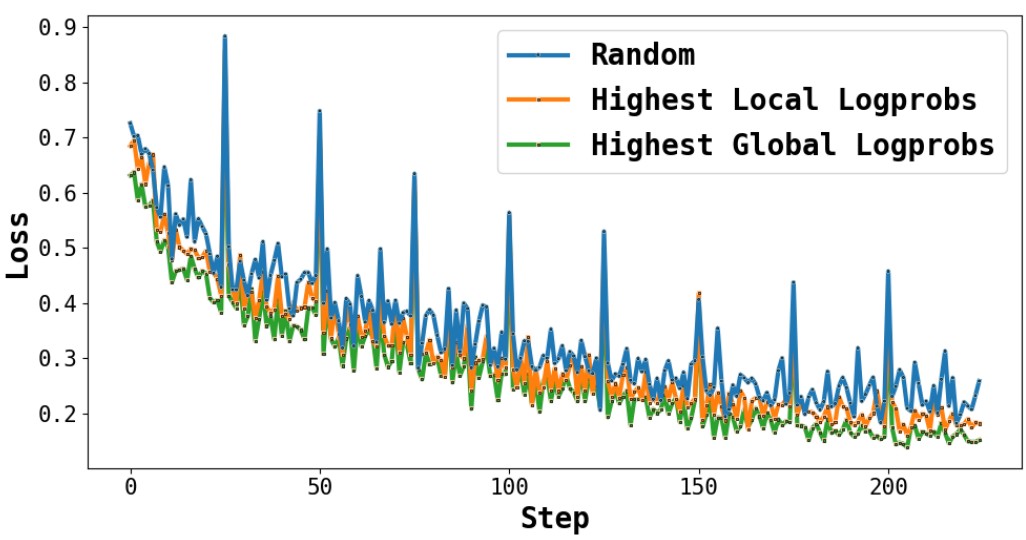

Figure 5: Loss plots of the student model Qwen2.5-32B-Instruct trained on randomly selected data points from LIMO responses, highest local log probabilities responses, and highest global log probabilities responses.

As shown in Figure 5, models trained on responses with the highest global log-likelihood demonstrate the fastest convergence and lowest training loss compared to those trained on randomly selected data or responses with the highest local log-likelihood. This behavior is expected, as high global log-likelihood responses likely represent more cohesive and natural samples as a whole, which align more closely with the student model's existing representation space. Such data may provide clearer learning signals, enabling the model to fit the training distribution more efficiently. However, as shown earlier in Table 3, instead, the model trained on data selected by highest local log-likelihood ultimately achieves better downstream performance. This highlights a key insight: while global log-likelihood data may facilitate faster convergence during training, this does not necessarily translate to better generalization, underscoring the limitations of relying solely on loss curves as indicators of final model performance.

## B.2 DATA COMPOSITION FROM SELECTION

In Section 4.2, we have chosen LIMO responses across three teachers (DeepSeek-R1, QWQ-32B, Qwen3-32B) based on local and global naturalness and provided results in Table 3. Here, we provide the composition of selected responses across teachers depending on the method in Table 7.

In Section 4.2, we selected LIMO responses from three teacher models, DeepSeek-R1, QWQ-32B, and Qwen3-32B, based on eihter local and global naturalness criteria, with the corresponding results presented in Table 3. In Table 7, we further detail the composition of the selected responses across teacher models for each selection method.

| | DeepSeek-R1 | QWQ-32B | Qwen3.0-32B |
|---|---|---|---|
| **Student: Qwen2.5-32B-Instruct** | | | |
| Random | 33.3 | 33.3 | 33.4 |
| Local Lowest | 42.4 | 11.3 | 46.3 |
| Global Highest | 47.6 | 7.2 | 45.2 |
| Local Highest | 42.4 | 36.3 | 21.3 |
| **Student: Qwen2.5-32B-Instruct** | | | |
| Random | 33.3 | 33.3 | 33.4 |
| Local Lowest | 43.3 | 20.4 | 36.3 |
| Global Highest | 47.2 | 8.6 | 44.2 |
| Local Highest | 26.8 | 44.9 | 28.3 |

Table 7: Data composition from different teacher models for the LIMO responses depending on the selection method(%).

## B.3 ADDITIONAL RESULTS ON MATH PROMPTS

We present additional results for Qwen2.5-7B-Instruct on the MATH benchmark, using responses generated by two different teacher models: Qwen2.5-72B-Instruct, which tends to produce shorter responses, and QWQ-32B, which generates longer reasoning responses. We provide a comprehensive summary of these results in Table 8.

| | | MATH | AIME25 | AMC | MINERVA | KAOYAN | OLYMPIADB | CN_MATH24 | AVG |
|---|---|---|---|---|---|---|---|---|---|
| | **Student: Qwen2.5-7B-Instruct** | | | | | | | | |
| | Original Model | 0.752 | 0.167 | 0.5 | 0.268 | 0.216 | 0.404 | 0.167 | 0.353 |
| **Teacher:** | Lowest LP | 0.678 | 0.1 | 0.3 | 0.224 | 0.296 | 0.314 | 0.133 | 0.292 |
| **Qwen2.5-72B** | Middle LP | 0.71 | 0.1 | 0.425 | 0.257 | 0.336 | 0.339 | 0.2 | 0.338 |
| **-Instruct** | Highest LP | 0.744 | 0.133 | 0.5 | 0.252 | 0.391 | 0.391 | 0.167 | 0.368 |
| **Teacher:** | Lowest LP | 0.686 | 0.1 | 0.4 | 0.246 | 0.286 | 0.324 | 0.2 | 0.320 |
| **QWQ-32** | Middle LP | 0.71 | 0.167 | 0.425 | 0.272 | 0.336 | 0.333 | 0.2 | 0.349 |
| | Highest LP | 0.732 | 0.167 | 0.475 | 0.279 | 0.412 | 0.382 | 0.233 | 0.382 |

Table 8: Performance of Qwen2.5-7B-Instruct student model on various reasoning tasks when fine-tuned with responses from different teacher models using MATH prompts. The log probabilities (LP) are categorized into lowest, middle, and highest based on the generation log probabilities of the student model.

## B.4 ABLATION: CONTEXT WINDOW SIZE VS PERFORMANCE

We provide an ablation study of the context window size in terms of performance. As we observe in Figure 6, the window size of 4 seems to be the optimal size for our case.

## B.5 GENERALIZABILITY EXPERIMENTS

To explicitly test this generalizability, we have since conducted additional experiments in general science. For the models trained on math data from our paper, we computed performance the GPQA-

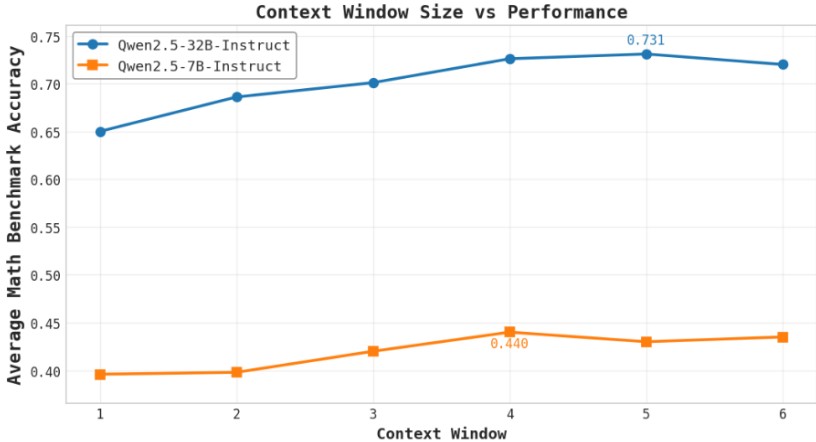

Figure 6: Context window size vs performance. Ablation on two student models.

Diamond benchmark, which tests expert-level reasoning across biology, physics, and chemistry. We obtained the following results:

Table 9: GPQA-Diamond Benchmark Results

| Model / Method | GPQA-Diamond (pass@1) |
|---|---|
| Original Qwen2.5-32B-Instruct | 0.551 |
| All 3 Teachers' Responses | 0.439 |
| LIMO-32B | 0.626 |
| Sky-T1-32B-Preview | 0.566 |
| OpenThinker2-32B | 0.646 |
| Global Highest (GRAPE) | 0.611 |
| Local Highest (Ours) | 0.702 |

Notably, our Local Naturalness selection method not only surpasses the Global Highest baseline but also outperforms other state-of-the-art models. This is particularly significant as these other models were trained on substantially larger and more diverse long reasoning datasets, underscoring the efficiency and effectiveness of our data curation technique.

To further demonstrate the generalizability of our approach, we extended our evaluation to code reasoning. We generated responses for 5,000 prompts from the OpenCodeReasoning and LeetCode datasets and used them to fine-tune the Qwen2.5-32B-Instruct model. The models performance were then evaluated on the LiveCodeBench v2 benchmark.

Table 10: LiveCodeBench v2 Benchmark Results

| Method | LiveCodeBench-easy | LiveCodeBench-medium | LiveCodeBench-hard |
|---|---|---|---|
| Original Qwen2.5-32B-Instruct | 0.890 | 0.471 | 0.114 |
| Global Highest (GRAPE) | 0.845 | 0.588 | 0.232 |
| Local Highest (Ours) | 0.874 | 0.633 | 0.261 |

As the results indicate, the student model trained on data selected via Local Naturalness consistently outperforms the one trained using the global log-probability baseline across the medium and hard difficulty tiers.

These additional results from the scientific and coding domains strongly suggest that the core principle of Local Naturalness is not confined to mathematics. The method's effectiveness in identifying high-quality reasoning data appears to generalize to other domains that require complex, step-by-step inference. We will include these findings in the paper to provide a more comprehensive evaluation of our method's applicability.

### B.6 Ablation: Llama-3.1-8B-Instruct

We provide results on another student model to show the generalizability of our method in Table 11.

| | MATH | AIME25 | AMC | MINERVA | KAOYAN | OLYMPIAD | CN_MATH24 | GPQA |
|---|---|---|---|---|---|---|---|---|
| | | | | Student: Llama-3.1-8B-Instruct | | | | |
| Original Model | 0.726 | 0.0 | 0.45 | 0.316 | 0.216 | 0.361 | 0.3 | 0.656 |
| Global Highest | 0.796 | 0.133 | 0.625 | 0.371 | 0.437 | 0.395 | 0.2 | 0.833 |
| Local Highest (Ours) | 0.814 | 0.167 | 0.725 | 0.368 | 0.432 | 0.410 | 0.2 | 0.879 |

Table 11: Performance of Llama-3.1-8B-Instruct student models on LIMO prompts when fine-tuned with responses from different selection strategies. The log probabilities (LP) are the global and local average log probabilities of the student model with non-greedy decoding with temperature 0.6, top-p 0.95 and over 8 samples, pass@8.

### B.7 Cross Teacher Selection: Performance with Non-Greedy Decoding

We provide results on cross teacher selection with non-greedy decoding in Table 12.

| | MATH | AIME25 | AMC | MINERVA | KAOYAN | OLYMPIAD | CN_MATH24 | AVG |
|---|---|---|---|---|---|---|---|---|
| | | | | Student: Qwen2.5-32B-Instruct | | | | |
| Original Model | 0.826 | 0.121 | 0.715 | 0.295 | 0.416 | 0.461 | 0.237 | 0.439 |
| All 3 Teachers | 0.835 | 0.400 | 0.887 | 0.335 | 0.578 | 0.559 | 0.583 | 0.597 |
| Global Highest | 0.862 | 0.442 | 0.903 | 0.338 | 0.629 | 0.634 | 0.721 | 0.647 |
| Local Highest (Ours) | 0.911 | 0.662 | 0.969 | 0.361 | 0.658 | 0.675 | 0.850 | **0.727** |

Table 12: Performance of Qwen2.5-7B-Instruct and Qwen2.5-32B-Instruct student models on LIMO prompts when fine-tuned with responses from different selection strategies. The log probabilities (LP) are the global and local average log probabilities of the student model with non-greedy decoding with temperature 0.6, top-p 0.95 and over 8 samples.

### B.8 Performance Comparison with Other SOTA Qwen-32B Models

We compare the performance of our model against several strong open-source implementations that also fine-tune the Qwen-32B-Instruct student model on comparable or larger datasets. LIMO-32B-V1(Ye et al., 2025) (available at `https://huggingface.co/GAIR/LIMO`) is trained on the LIMO prompt set using responses exclusively from the DeepSeek-R1 teacher. Sky-T1-32B-Preview(Team, 2025a) (available at `https://huggingface.co/NovaSky-AI/Sky-T1-32B-Preview`) is trained on a 17K example dataset (`https://huggingface.co/datasets/NovaSky-AI/Sky-T1_data_17k`) generated using the QWQ-32B model. OpenThinker2-32B(Team, 2025b) (available at `https://huggingface.co/open-thoughts/OpenThinker2-32B`) is trained on a substantially larger dataset of 1.04M samples(`https://huggingface.co/datasets/open-thoughts/OpenThoughts2-1M`), also generated using DeepSeek-R1 as the teacher. A detailed comparison of the results is provided in Table 13.

| | MATH | AIME25 | AMC | MINERVA | KAOYAN | OLYMPIAD | CN_MATH24 | AVG | GPQA |
|---|---|---|---|---|---|---|---|---|---|
| | | | | Student: Qwen2.5-32B-Instruct | | | | | |
| Original Model | 0.824 | 0.133 | 0.700 | 0.298 | 0.422 | 0.471 | 0.233 | 0.445 | 0.551 |
| Global Highest | 0.876 | 0.433 | 0.825 | 0.331 | 0.592 | 0.636 | 0.733 | 0.632 | 0.611 |
| LIMO-32B | 0.896 | 0.433 | 0.925 | 0.346 | 0.618 | 0.630 | 0.800 | 0.664 | 0.626 |
| Sky-T1-32B-Preview | 0.876 | 0.200 | 0.750 | 0.301 | 0.558 | 0.507 | 0.533 | 0.532 | 0.566 |
| OpenThinker2-32B | 0.922 | 0.567 | 0.900 | 0.324 | 0.648 | 0.640 | 0.833 | 0.691 | 0.646 |
| Local Highest (Ours) | 0.902 | 0.667 | 1.000 | 0.353 | 0.653 | 0.673 | 0.833 | **0.726** | 0.694 |

Table 13: A performance comparison of our model with other open-source SOTA models fine tuned on the Qwen2.5-32B-Instruct student model.

All experiments, including sampling, training, and evaluation, were conducted using either 4×H100 GPUs or publicly available APIs when applicable. Upon completion of the review process, we are committed to releasing our code, datasets, and trained models to support and accelerate further research within the community.

