# OpenReview forum: "Distilling Reasoning into Student LLMs: Local Naturalness for Selecting Teacher Data"
_ICLR.cc/2026/Conference — ICLR 2026 Conference Withdrawn Submission_

### Official Review · Reviewer_WBPs · 2025-10-16

**Soundness:** 2
**Presentation:** 1
**Contribution:** 2
**Rating:** 2
**Confidence:** 3

**Summary:**

The paper proposes a scoring method for selecting teacher outputs for a given prompt that were generated using different models, which uses an average of probabilities of reasoning steps instead of a global average, which is used in a previous work.
The authors show that this kind of acquisition function is more reliable than global (i.e. sequence-level) scores when used for longer reasoning chains.

**Strengths:**

- The presented methods shows improvements over prior methods

- The method itself is intuitive and easy-to-implement

**Weaknesses:**

- The work is rather incremental (which can be okay in itself) and more of a tweak of a previous method called GRAPE.

- The paper could discuss related works better. For one, I don't think it is fair to say that curriculum / reweighting approaches necessitate complicated training loop modifications or that using a surrogate model to predict data "goodness" is too expensive. The first could involve simple scaling in a weighted-loss which is easy to implement, similar to Thakkar et al. 2023, and the latter should clearly be cheaper (in terms of runtime) than the proposed strategy if the surrogate model is smaller than the student model.
For another, the method is very similar to active learning research, where data is often selected according to information-theoretic or uncertainty-based measures which could easily be adapted to this setting (although there it is often assumed that no labels are available). However, this is only very briefly discussed and no such baselines are used. At the same time, the distinction between selecting inputs and input, output pairs could be made more clear.

- The writing could be improved and made more precise, at times the paper is quite verbose. Separating big blocks of text into paragraphs would also help. The paper also mixes methodology and experiments a lot which makes it hard to compare methods quickly and makes it necessary to jump a lot within the paper.

- Transposing the axes from Fig. 2 (x: log-probs, y: accuracy) to Fig. 3 (x: accuracy, y: log-probs) is very confusing, is there any reason for choosing this?

- I think some more simple baselines would help the paper significantly. For example, one could draw a model at random and then take the output with highest log probability to compare to? In general, it would be good to discuss trade-offs in runtime, for example, the fully random picking already performs quite well but does not need additional forward passes for selection which is costly. There could be a baseline where a random set of examples is used which matches the training time of your method. Without this axis comparison is not entirely fair.

**Questions:**

- Please make your figures .pdfs.

- "Recognizing that a one-size-fits-all approach to SFT data is suboptimal" (L108) what does this mean?

- L116 and 118 have one set of parentheses too many

- The notation mixes bold x / y and unbolded x / y for full sequences.

- Table 2 is missing whitespaces

## References

Megh Thakkar, Tolga Bolukbasi, Sriram Ganapathy, Shikhar Vashishth, Sarath Chandar, and Partha Talukdar. 2023. Self-Influence Guided Data Reweighting for Language Model Pre-training. In Proceedings of the 2023 Conference on Empirical Methods in Natural Language Processing, pages 2033–2045, Singapore. Association for Computational Linguistics.

---

### Official Review · Reviewer_UEVF · 2025-10-31

**Soundness:** 2
**Presentation:** 1
**Contribution:** 2
**Rating:** 2
**Confidence:** 4

**Summary:**

This paper proposes Local Naturalness, a student-aware metric for selecting teacher responses in reasoning distillation, aiming to replace global log-probability scoring. The idea is interesting and relevant, showing some empirical improvements. However, the paper suffers from unclear writing, limited analysis, and weak comparisons. It lacks theoretical insight, broader validation, and discussion of computational cost.

Refer to the detailed comments below.

**Strengths:**

1. The paper identifies a genuine and important problem: the failure of "global naturalness" (full-sequence log-probability) for data selection in multi-teacher, long-reasoning scenarios.

2. The experiments cover multiple student–teacher pairs (Qwen, DeepSeek, Gemma) and several reasoning datasets.

3. Reproducibility. Appendices list datasets, model names, and hyperparameters clearly.

**Weaknesses:**

1. Writing and Structure: The paper is poorly written and organized. For example, sections 3.2/3.3 aim to explain when and why global log-probability fails, but do so unclearly. Excessive implementation details (e.g., hyperparameters) distract from the main idea, and many figures/tables are confusing or mislabeled. Some captions (e.g., fig 2) appear incorrect and make the results misleading.

2. Lack of Analysis and Insight: Local Naturalness is presented as an empirical heuristic without analytical justification or deeper understanding of why it works. The paper provides lengthy experimental descriptions but little conceptual insight. It shows that the method improves performance but does not convincingly explain the underlying reason.

3. Computational Cost: The proposed method requires computing log-probabilities for every sentence in each response using a sliding context window, which is far more expensive than e.g., global scoring. Although claimed to be cheaper than some offline methods, computational cost and comparison with baselines are worth including.

4. All results are restricted to mathematical reasoning tasks. No evidence is shown that the method generalizes to other reasoning or coding domains.

5. The paper primarily compares against global naturalness and random selection. Comparisons with other widely used approaches, such as self-consistency, which are commonly applied in reasoning data selection, should be considered.

**Questions:**

1. how to choose the context window k? The choice of k=4 is justified by an ablation study in the appendix, but the reasoning behind why this particular size is optimal is not explored. Is it related to the average number of sentences per reasoning step in the data? Also, looks like the two student models used in the ablation study is not very consistent, and I would like to see the trend after k=6

2. In table 6, it is interesting to see that local highest performs much better than random, while local lowest is only slightly worse. Could the authors explain why this asymmetry occurs and provide evidence?

---

### Official Review · Reviewer_ySeL · 2025-10-31

**Soundness:** 2
**Presentation:** 3
**Contribution:** 3
**Rating:** 4
**Confidence:** 3

**Summary:**

The paper studies response selection for reasoning distillation when multiple teacher models provide multiple long CoT traces per prompt. It shows that the widely used global log-probability (student’s average next-token log-likelihood over the whole response) correlates poorly with downstream performance in mixed-teacher, long-context settings. The authors propose Local Naturalness: score each response by averaging sentence-level log-probabilities conditioned on a small sliding window (k previous sentences), then aggregate across steps. Two applications are demonstrated: (i) teacher selection (rank teachers by student-assessed average local scores) and (ii) mixed-teacher response selection (pick the locally highest-scored response per prompt). Across Qwen/Llama students and several strong teachers, local scoring reliably identifies better teachers and yields superior SFT sets compared to global scoring, random, and even training on the single best teacher’s full set. Ablations on window size and analyses on convergence vs generalization are provided; extensions to GPQA-Diamond and LiveCodeBench suggest broader applicability.

**Strengths:**

The paper addresses a highly practical and underexplored setting: multi-teacher, long-CoT distillation, where response selection is critical and nontrivial. The empirical finding that global log-probability is unreliable across teachers for long traces is important and well-supported with careful experiments. Local Naturalness is simple, training-free, and student-aware, integrates easily into SFT pipelines, and improves both teacher choice and per-prompt selection in a robust way. The results are consistent across student sizes and evaluation suites; notably, locally curated mixed-teacher data can surpass the single best teacher’s dataset. The analysis connecting faster training loss (global) versus better generalization (local) is insightful, and the ablation on window size provides actionable guidance. Additional experiments on GPQA-D and LiveCodeBench strengthen claims of broader applicability.

**Weaknesses:**

The method’s design choices around step segmentation (sentence-based) are not deeply examined; math/science reasoning includes formulas and multi-line derivations where sentence boundaries may be suboptimal. A comparative study against equation/LaTeX-block or PRM-informed step boundaries could strengthen the case. The mechanistic explanation for why global fails in long contexts is mostly intuitive (long-context degradation); adding quantitative evidence (per-length perplexity trends, attention entropy/retention, KV cache effects) would be compelling. Compute overhead is not fully quantified: while local windows are shorter than full-context scoring, sentence-wise scoring across large corpora can still be expensive. A cost–benefit analysis (accuracy gain per GPU-hour) and comparisons to verifier-based selectors (PRM/ORM) would help practitioners. Finally, related work comparison is incomplete: recent “one-shot CFT” (critique-focused distillation) also operates in multi-teacher, multi-response settings (multiple teachers generate critiques for the same prompt). A systematic citation and empirical comparison would situate the contribution more clearly.

**Questions:**

How sensitive are gains to the definition of a “step”? Have you compared sentence-based segmentation with (i) equation/LaTeX block segmentation, (ii) PRM-driven step boundaries, or (iii) clause-level segmentation using discourse markers? Even a small ablation could reveal more robust segmenters for math/science. Can you provide mechanistic evidence for long-context degradation that hurts global scoring—e.g., correlation of downstream accuracy with response length strata (1k/5k/10k/20k), changes in attention entropy or retrieval failure across depth, or KV-cache utilization patterns? What is the end-to-end compute cost to compute local scores at the corpus scale versus global scoring and versus PRM-based selection? Under equal compute budgets, how do local scoring, PRM/ORM reranking, and hybrid methods (local+PRM) compare on accuracy? Beyond math/science/code, have you tried natural language multi-hop reasoning (HotpotQA, StrategyQA)? Do local scores still outperform global when steps are less formulaic? Does Local Highest induce teacher skew (data diversity collapse)? Would selecting top-2 local responses per prompt or adding a diversity regularizer help further?

Please add and compare against the recent “one-shot CFT” that likewise evaluates multiple teacher outputs for the same prompt (e.g., multi-teacher critiques for a single problem). It is a close setting to yours—multiple teachers, multiple responses, selection/fusion for SFT.

---

### Official Review · Reviewer_JAJQ · 2025-11-08

**Soundness:** 3
**Presentation:** 3
**Contribution:** 2
**Rating:** 4
**Confidence:** 4

**Summary:**

This paper proposes a new data selection method for distilling reasoning abilities from teacher LLMs to student LLMs. Specifically, this paper focuses on response-centric data selection, that is, selecting better responses from multiple teachers. The authors design a new metric, named local naturalness, which scores a response by measuring the student’s log-probabilities over short, sequential reasoning steps (e.g., sentences) conditioned only on a small local window. Compared to the previous widely-used global log-probability (i.e., global "naturalness"), this new metric can select responses aligning better with the student capacities. The authors conduct extensive experiments using various teacher LLMs and student LLMs. The experimental results demonstrate the effectiveness of the proposed method.

**Strengths:**

- The topic of selecting more effective post-training data for distillation is meaningful, especially as the paper focuses on data selection from the perspective of responses.
- The proposed method, local naturalness, is concise, easy to implement, and cost-efficient, showing potential for large-scale application.
- The analysis is insightful. Through well-controlled experiments on single-teacher distillation, the paper effectively investigates the relationship between local naturalness scores and model performance, demonstrating that local naturalness exhibits a higher correlation with task performance compared to global naturalness.
- The presentation is clear and easy to follow. The paper first analyzes the effectiveness of local naturalness and then applies it for data selection and model training.

**Weaknesses:**

- Regarding why local naturalness works well, I believe additional analysis is needed. Compared to global naturalness, does local naturalness capture more fine-grained reasoning information? I appreciate the simplicity of the proposed approach, but the paper would benefit from either a theoretical justification or a more comprehensive empirical analysis.
- I think the main weakness of the paper lies in the experimental setup. The experiments are conducted only on LIMO and another dataset with around 8,890 samples. However, in realistic post-training scenarios, we often use tens or even hundreds of thousands of samples. It remains unclear whether the proposed data selection method would still be effective under such large-scale conditions. I suggest the authors either validate local naturalness on a larger training set or demonstrate its data efficiency. For example, using local naturalness selection, how many times more tokens are needed to achieve comparable performance without this selection method?

**Questions:**

Please see the weaknesses above.

---

### Note · Authors · 2025-11-19

**Comment:**

We appreciate the feedback and will improve the work in the next revision.

**Withdrawal Confirmation:**

I have read and agree with the venue's withdrawal policy on behalf of myself and my co-authors.